# The Vagal Nerve, Inflammation, and Diabetes—A Holy Triangle

**DOI:** 10.3390/cells12121632

**Published:** 2023-06-15

**Authors:** Liat Sorski, Yori Gidron

**Affiliations:** 1Sagol School of Neuroscience and School of Psychological Sciences, Tel Aviv University, Tel Aviv 6997801, Israel; liatsors@tauex.tau.ac.il; 2Department of Nursing, Faculty of Social Welfare and Health Sciences, Haifa University, Haifa 3498838, Israel

**Keywords:** inflammation, vagus nerve, heart rate variability (HRV), type 2 diabetic mellitus (TDM2)

## Abstract

Type 2 diabetic mellitus (T2DM) is a common chronic disease and a substantial risk factor of other fatal illnesses. At its core is insulin resistance, where chronic low-level inflammation is among its main causes. Thus, it is crucial to modulate this inflammation. This review paper provides scientific neuroimmunological evidence on the protective roles of the vagal nerve in T2DM. First, the vagus inhibits inflammation in a reflexive manner via neuroendocrine and neuroimmunological routes. This may also occur at the level of brain networks. Second, studies have shown that vagal activity, as indexed by heart-rate variability (HRV), is inversely related to diabetes and that low HRV is a predictor of T2DM. Finally, some emerging evidence shows that vagal nerve activation may reduce biomarkers and processes related to diabetes. Future randomized controlled trials are needed to test the effects of vagal nerve activation on T2DM and its underlying anti-inflammatory mechanisms.

## 1. Basic Epidemiology of Diabetes Mellitus

Diabetes-mellitus is a chronic disease that alters the way the body uses glucose, resulting, among other things, in inadequate management and processing of glucose blood levels. There are two main subclassifications of diabetes-mellitus, namely, type 1 and type 2.

The insulin-dependent type 1 diabetes mellitus (T1DM) is an autoimmune disorder in which insulin-secreting beta cells in the pancreas are destructed by T lymphocytes [1,2]. This leads to insulin deficiency due to the inability of the pancreas to produce it. The onset of T1DM pathogenesis could be triggered by genetic and environmental factors, and its rate of development is rapid on the occasion of juvenile-onset in youths, or gradual, on the occasion of late onset in adults [3].

Type 2 diabetes mellitus (T2DM), which is non-insulin dependent, is also characterized by deficient insulin; however, this type is triggered by a malfunctioned interplay between insulin blood levels and cells’ insulin insensitivity. Specifically, the main role of insulin is to act as a vector for blood glucose to enter cells. However, in T2DM, beta cells are compromised due to the intensive pressure to produce insulin, aiming to lower the elevated glucose levels in the blood because of cells’ resistance to insulin [4]. This overburden leads to beta cells’ malfunctioning. T2DM pathogenesis is mainly triggered by obesity and aging, but it is well established that it also has a greater hereditary basis compared to T1DM [5]. T2DM onset usually occurs during adulthood, even though the increased rate of obesity seen today in youth has caused the incidence of T2DM to rise in younger ages as well [6].

In both types of diabetes, the loss of beta cells leads to hyperglycemia due to insulin deficiency, with the need for permanent insulin therapy or adjustment, to sustain a functioning glycemic balance [7].

According to the CDC (centers for disease control and prevention), 37.3 million people, which is 11.3% of the US population, have diabetes, of whom 28.7 million are diagnosed and 8.5 million are not diagnosed. T1DM comprises 5–10% of all the incidents of diabetes, and T2DM constitutes 90% of the incidents.

## 2. The Role of Inflammation in DM

Inflammation is the activation of immunological responses generated by the body to protect itself from pathogens (e.g., bacteria and toxins), and to promote tissue repair and recovery. When it is mild and time limited, this response is protective. However, during the past decades systemic chronic inflammation has been shown to contribute largely to the onset and progression of major fatal illnesses, including cancer, heart disease, diabetes, and neurodegenerative diseases [8]. Inflammation has a significant role in the development and progression of diabetes. It was shown that chronic low-grade inflammation had been associated with insulin resistance, a known characteristic of T2DM [9]. Inflammation promotes the activation of pro-inflammatory cytokine signals which block beta-cells’ insulin receptors of pancreatic islets, thus preventing the effectual response of the body to insulin [10]. Inflammation contributes to this process by affecting the way insulin signals are transmitted within the cells, making it harder for the body to respond to insulin effectively, an outcome called insulin resistance, a hallmark of T2DM.

Inflammation, oxidative stress and protein kinase C (PKC) activation, are known to trigger a decreased activity of insulin receptor substrate 1 (IRS1) through increased phosphorylation of its serine/threonine and reduced phosphorylation of tyrosine, all of which were found as compromising the signaling of insulin [11]. Additionally, there is a linkage between inflammation and insulin resistance, with the advanced glycation end-product AGE/RAGE/NF-kB axis being at the core of this pathological process [12].

Another key actor in the development of both inflammation and insulin resistance is TLR4, which is expressed in insulin target tissues. In the context of obesity, a risk factor for T2DM, TLR4 causes insulin resistance and metabolic inflammation through the upregulation of proinflammatory genes’ transcription and the activation of the proinflammatory kinases JNK, IKK, and p38 [13].

Inflammation is also associated with the promotion of diabetes-related complications, such as cardiovascular disease and neuropathy [8,14]. As chronic inflammation can increase the risk of developing such complications, it is crucial to manage both diabetes and inflammation effectively.

## 3. The Role of the Vagus Nerve in DM: Epidemiology and Clinical Studies

The vagal nerve is the 10th cranial nerve and a major branch of the parasympathetic nervous system. At the present time, the best non-invasive manner for measuring vagal nerve activity is by measuring heart-rate variability (HRV), the fluctuations in the intervals between normal heartbeats. Indeed, HRV is profoundly strongly correlated with actual vagal nerve activity (r = 0.88) [15]. One indirect way to examine the role of the vagus in T2DM is via linking psychological stress with T2DM since stress is inversely correlated with HRV, based on a review of 37 studies [16]. In a unique longitudinal study with 12,844 Australian women initially free of diabetes, those reporting moderate-high stress levels had a 2.3 times higher risk of developing later diabetes. Furthermore, this association remained significant after controlling for factors such as hypertension, smoking, etc. [17].

What is the evidence linking vagal nerve activity with diabetes? A meta-analysis of 25 case–control studies with 1356 diabetic patients and 1576 healthy controls compared those groups on HRV. This review revealed that multiple HRV time domain parameters (e.g., SDNN and RMSSD) and frequency domain parameters (e.g., high-frequency (HF) and low-frequency (LF) HRV) were significantly lower in diabetic patients than in healthy controls. In contrast, blood glucose levels were significantly associated with higher levels of several HRV parameters (e.g., HF-HRV, root mean square of the successive differences (RMSSD)) [18]. In a unique longitudinal study on 9192 Brazilians, lower HRV (Standard deviation of the NN (R-R) intervals (SDNN), RMSSD, the proportion of NN intervals larger than 50msec divided by the total number of NN (R-R) intervals (pNN50), HF-HRV, and LF-HRV) significantly predicted a higher risk for developing diabetes [19]. Finally, though not an experimental study, one study compared diabetic patients with controlled glucose to diabetic patients without controlled glucose and to healthy controls on HRV. That study found that HRV was lower in diabetic patients with poor glycemic control than in diabetic patients with good glycemic control, suggesting that a more severe illness may also result or even cause poorer vagal nerve activity [20]. One mechanism of this relation may be due to impairments found in the lungs due to diabetes [21], reducing respiration capacity, which may easily affect one’s HRV.

## 4. The Anti-Inflammatory Effects of Vagal Nerve Activation

A recent article elegantly reviewed the scientific literature on the topic of vagal modulation of inflammation [22], which is termed the cholinergic anti-inflammatory pathway. The basis of this body of knowledge derives from the scientific domain of neuroimmunology and its sister domain, psychoneuroimmunology. Several neurobiological pathways exist through which the vagal nerve modulates and inhibits excessive inflammation. First, vagal nerve paraganglia express receptors for peripheral interleukin-1 (IL-1; [23], a major inflammatory cytokine that may perform a crucial role in the progression of T2DM [24]. Then, several anti-inflammatory routes are activated to control peripheral inflammation. The first route is a central neuro-hormonal route including the hypothalamic-pituitary-adrenal (HPA) axis. This occurs after afferent vagal conversion of the peripheral IL-1 signal to acetylcholine, which then activates the hypothalamus. This triggers the HPA axis to secrete cortisol, which obviously has an anti-inflammatory effect [25]. The second route is an efferent neuro-immunological one, in which efferent vagal fibers reach the celiac ganglion and convert to a sympathetic branch that innervates the spleen. In the spleen, a sub-group of “cholinergic T-cells” have been identified, namely CD4+ CD44high CD62low Chat EGFP+ cells [26]. These special T-cells, as other T-cells, express beta-adrenergic receptors for receiving sympathetic signaling. Once the beta-adrenergic receptor of these “cholinergic T-cells” is signaled, they produce the vagal neurotransmitter acetylcholine (Ach), which then reaches its receptor alpha-7-nicotinic Ach receptor on splenic macrophages. This macrophage signaling ultimately results in the reduced synthesis of pro-inflammatory cytokines [26]. However, additional routes may exist, through which the vagal nerve modulates inflammation which includes possible direct innervation of the spleen by the vagus, vagal stimulation of intestinal segments, and vagal stimulation of the adrenal gland resulting in secretion of dopamine [27]. Future research needs to clarify those additional routes.

These pathways may also be taken to a higher, cortical level. There is a positive correlation between HRV and the connectivity between the prefrontal cortex and the amygdala [28]. Furthermore, a similar connectivity between the ventromedial prefrontal cortex and the amygdala, is inversely related to peripheral inflammation [29]. Therefore, neuromodulation of inflammation may involve frontal-limbic brain connectivity, the vagus nerve and peripheral inflammation, all of which may potentially reduce insulin resistance.

A multitude of correlational studies have examined relationships between HRV, the vagal nerve index, and peripheral inflammation. These were reviewed in a meta-analysis by Williams et al. [30]. This review found quite consistent evidence for an inverse correlation between the HRV parameters of HF-HRV and RMSSD with CRP (C-reactive protein), white blood cell count, and fibrinogen, and between HF-HRV and IL-6 [30]. In a large-scale study in which confounders were also considered, HF-HRV was inversely correlated with CRP and fibrinogen, after statistically adjusting for the effects of many demographic and lifestyle confounders (e.g., age, ethnicity, and smoking) and disease history [31]. Marsland et al. (2007) found that levels of HRV after performing paced breathing were inversely correlated with IL-1b, IL-6, and TNF in whole blood stimulated by LPS [32]. These results are important for two reasons—first, they used ex vivo functional levels of reactive inflammation, and second, the study measured HRV after vagal activation by paced breathing. These steps provide evidence for the inverse relation between HRV and inflammation after inducing dynamic neuro-immunomodulation. Since these observations were found ex vivo, without concurrent parasympathetic influences, they suggest a programming effect of vagal activation on immune cell production of inflammatory signals [33].

Experimental studies in humans found that vagal nerve activation reduces peripheral inflammation. We review here only the studies which used a control group. In a study on patients with migraine, half received transcutaneous vagal nerve stimulation (tVNS) by the Gammacore device placed on the neck, and half received sham stimulation, for 120 s/day, for 2 months. Only IL-1, but not IL-6, TNF and IL-10, significantly decreased in the tVNS group to a greater extent than in controls [34]. In another study which included LPS-induction of inflammation, 10 participants received tVNS while 10 others received sham stimulation, both including 2 stimulations of 4 min each. In the tVNS group, IL-8 levels declined from baseline after 90 min more than in controls. Furthermore, after 24 h, those who received tVNS showed greater reductions in IL-1b, IL-8, and TNF and higher increases in the anti-inflammatory cytokine IL-10, than controls [35]. Together, these studies show biologically plausible mechanisms (HPA-axis, a splenic route) and empirical evidence for the inhibitory effects of the vagal nerve on inflammation.

Beyond these important anti-inflammatory effects of the vagus, this nerve has many more effects, which place it as a factor of potentially greater importance than other risk factors of diabetes. Activating the vagus helps to reduce eating in people with high cravings for food [36] and HRV increases with performing more physical activity, such as resistance exercise, particularly in middle-aged people [37]. This is crucial for preventing T2DM. The robust and comprehensive bi-directional effects of the vagal nerve on behavioral lifestyle factors (diet, exercise, and smoking) and on biological mediators (oxidative stress, inflammation, and sympathetic hyperactivity) have been proposed in a neuroimmunological framework of fatal diseases [38], which indeed proposes the centrality of this nerve in disease prediction and prevention.

## 5. Effects of Vagal Nerve Activation in Diabetes

### 5.1. Effects of Vagal Nerve Stimulation in Diabetes

In contrast to many studies which examined relations between HRV and DM, far fewer studies examined the effects of vagal nerve activation on clinical outcomes in DM. These include experiments in animals and HRV-biofeedback in patients with DM, which we shall now explain. In animals, vagal stimulation may include vagomimetic activation at the cellular level via drugs and electrical vagal nerve stimulation. In an experimental study performed on Sprague Dawley rats, celiac vagal transection reduced beta-cell proliferation by 50%. These results suggest that the vagus may strongly influence pancreatic beta-cell proliferation which could be crucial during times of elevated requirements for insulin [39]. In another study performed on isolated beta cells, muscarinic (vagal) stimulation led to increases in glucose-induced beta-cell proliferation, while adrenergic stimulation had the opposite effect [40]. One study induced an animal model of T2DM and tested the effects of an implanted efferent vagal nerve stimulation (eVNS) on glycemia. eVNS reduced glycemia consistently, and a certain dose was even found in these rats to be the most effective [41].

Another important study mimicked the effects of transcutaneous auricular vagal nerve stimulation (taVNS) in diabetic rats and compared it to a control stimulation. The taVNS led to reduced body weight, lower levels of blood glucose levels, and less depressive-like behavior compared to controls [20]. Additionally, vagus nerve stimulation is associated with improved glucose metabolism and insulin secretion. For instance, it has a significant impact on GLP, a gastrointestinal hormone with an important role in glucose homeostasis, by mediating the regulatory effects of its peripheral and central production on metabolism [42]. However, studies have shown inconsistent findings, with some reporting improved glucose control following VNS while others show no such beneficial effect. One study compared the effects of VNS with pharmacotherapy versus pharmacotherapy alone in patients with epilepsy, in whom implanted VNS has been approved for a few decades. No differences in blood glucose levels were found between both groups. However, the timing of VNS was important since stimulating with long on and short off periods led to increases in glucose, while stimulating with short on and long off periods appeared to have no effect or even decreased glucose levels [43]. Though only this study examined this issue, more research needs to confirm these findings in people with epilepsy or T2DM and in healthy people.

### 5.2. Effects of Vagal Nerve Biofeedback on Diabetes

Human studies on vagal nerve activation in DM are unfortunately scarce. One pilot study tested the effects of HRV-biofeedback in DM patients. In this treatment, patients learn to perform paced slow breathing to increase their HRV, while receiving biofeedback on a screen for their performance. In that study, levels of HbA1C decreased by 1.3%, weight decreased by 4.0 kg, SBP decreased by 8.6 mmHg and plasma fasting glucose decreased by 4.3 mmol/L [44]. However, no control group was included.

## 6. Conclusions and Future Directions

Recently, there has been an emerging interest in using HRV as a potential risk marker for developing diabetes and an increased interest in using HRV-biofeedback as a promising technique for possibly managing DM. As cardiovascular disease is a common complication of diabetes, using HRV to monitor the progression of diabetes would be beneficial, as diabetic patients tend to present with reduced HRV, which is also associated with a higher risk for cardiovascular disease [45,46] and for post-MI mortality [47].

Another potential therapeutic use of HRV in the context of diabetes is HRV-based interventions to improve the regulation of glucose blood levels. For example, it was shown that HRV-based interventions, such as yoga or exercises which involve deep breathing, can improve HRV and glucose regulation in T2DM patients [48,49,50]. Integration of such interventions into diabetes treatment plans will enable a potential reduction in the risk for complications accompanying diabetes. However, the effects of HRV-biofeedback on T2DM must be tested in randomized controlled trials (RCT) to examine the causal and clinical effects of vagal activation in this condition.

HRV can also be utilized as a predictor of medical complications that are related to diabetes, such as heart disease, neuropathy, chronic kidney disease, nerve damage, and additional problems involving the feet. For instance, HRV has been shown to be a marker of autonomic dysfunction, which is a common complication of diabetes [46,51]. Measuring HRV to monitor patients for the development of autonomic dysfunction will enable early intervention and possibly prevention or at least a reduction of the severity of such complications.

However, it is crucial to keep in mind that the relationship between the vagal nerve and DM may be bi-directional. Indeed, some of the experimental and prospective studies reviewed above propose that the vagal nerve’s functioning has a causal role in T2DM, but some of the evidence may also suggest that T2DM itself may impair vagal nerve activity, indexed by low HRV in patients with poor glycemic control. It is especially possible that neuropathy can impair vagal activity, however, since many studies conceptualize neuropathy as low HRV, other ways to measure neuropathy need to be used to examine this issue.

In conclusion, the vagal nerve seems to be an essential regulatory factor in managing diabetes and there are several directions for its use as a potential future therapeutic target. However, more research is needed to confirm these findings and to better understand the underlying mechanisms. Vagal nerve stimulation by taVNS, HRV biofeedback, and HRV-based monitoring are a few of the promising future directions for the inclusion of the vagal nerve as a potential treatment target in diabetes. Figure 1 depicts our proposed model of vagal nerve modulation of diabetes-related inflammation.

## Figures and Tables

**Figure 1 cells-12-01632-f001:**
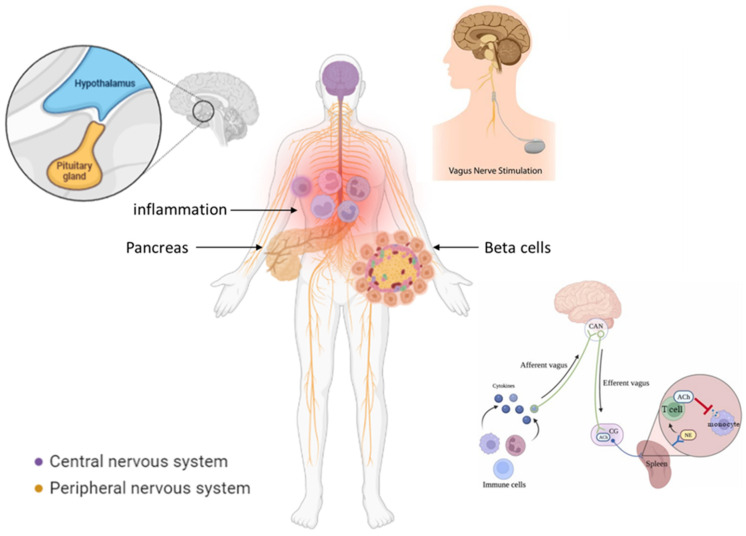
Vagal modulation of inflammation in diabetes. The vagal nerve can regulate factors that are crucial for diabetes: 1. It influences beta-cells’ proliferation in the pancreas. 2. It inhibits systemic inflammation via activation of the HPA-axis and via vagal—sympathetic innervation of the spleen, where certain T-cells send cholinergic signals to local macrophages to inhibit inflammatory cytokines. The figure was created with BioRender.com (accessed on 2 March 2023).

## Data Availability

No new data were created or analyzed in this study. Data sharing is not applicable to this article.

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
