# Peer review of "The Vagal Nerve, Inflammation, and Diabetes—A Holy Triangle"

_cells, 2023, doi:10.3390/cells12121632_

Round 1
Reviewer 1 Report
The commentary submitted by Liat Sorski and Yori Gidron aimed to provide scientific evidence on the protective roles of the vagal nerve in T2DM. The authors highlighted that vagal activity indexed by heart-rate variability (HRV) is inversely related to diabetes and low HRV predicts T2DM. They have concluded that randomized controlled trials are needed to test the effects of vagal nerve activation on T2DM and its underlying anti-inflammatory mechanisms. The commentary posed by the authors is well-defined and the manuscript adheres to the relevant standards for reporting and data deposition. However, few suggestions to improve the writing.
It would be better to highlight the Pathological changes in the vagus nerve in diabetes separately.
The Strategies of Vagal Nerve Stimulation can be given separately.
The acknowledgments section needs to be added correctly.
Grammatical errors should be fixed throughout the manuscript.
Grammatical errors should be fixed throughout the manuscript.
Author Response
Reviewer 1:
Thank you for your notes and thoughts.
Q1.
Effects of DM on vagal functioning need to be mentioned. We thank the reviewer for this important comment. We thus added the following:
Finally, though not an experimental study, one study compared diabetic patients with controlled glucose with diabetic patients without controlled glucose and health con-trols, on HRV. That study found that HRV was lower diabetic patients with poor glycemic control than in those with good glycemic control, suggesting that a more severe illness may also result or even cause poorer vagal nerve activity (Yu et al., 2020; 46). One mechanism of this relation may be due to impairments found in lungs due to dia-betes (Pitocco et al., 2012), reducing respiration capacity, which may easily affect one's HRV.
Q2.
We separated the 2 parts of vagal nerve activation into effects via vagal nerve stimulation (section 5.1) and effects of vagal nerve biofeedback in diabetes (section 5.2).
Q3. We added a proper sentence of acknowledgements.
Q4. The English has been corrected.
Reviewer 2 Report
The authors adequately review the interesting topic of the role of the vagal nerve in inhibiting neuroinflammation. They discuss the utility of HRV as an index of vagal activity and the observation that low HRV is a predictor of T2DM. The potential of using vagal nerve activation to benefit patients with T2DM is also discussed.
Did the authors look for any studies on epilepsy patients with and without DM population that are using VNS treatment (which has been available for several decades)?
Additional Comments:
Abstract: Well-written
Manuscript:
1. Basic epidemiology of DM: Sufficiently covered
2. Role of inflammation in DM: Sufficient. Many abbreviations are used without the full meaning (PKC, JNK, IKK etc.). Should provide a list of abbreviations with an explanation at the end of the manuscript)
3. Role of vagus-----: Well written
4. Anti-infla-----: Well written
5. Effects of vagal nerve------: Well written
Conclusion: Major problem is that abnormal HRV in diabetic patients may simply be the result of diabetic autonomic neuropathy which will make the author's conclusions questionable. There should be a discussion of this possibility.
Only minor corrections needed to make some of the sentences shorter and more precise.
Author Response
Reviewer 2:
Thank you for your notes and comments.
Q1. Effects of vagal nerve stimulation (VNS) on diabetes in patients with epilepsy. We thank the reviewer for this important comment. We added these sentences:
One study compared effects of VNS with pharmacotherapy versus pharmacotherapy alone in patients with epilepsy, in whom implanted VNS has been approved for a few decades. No differences in blood glucose levels were found between both groups. However, the timing of VNS was important – stimulating with long on and short off periods led to increases in glucose, while stimulating with short on and long off periods appeared to have no effect or even decreased glucose levels (Stauss et al., 2019). Though only this study examined this issue, more research needs to confirm these findings in people with epilepsy and in health people.
Q2. Abbreviations will be explained.
Q3. The possibility that diabetic neuropathy may cause poor vagal nerve activity and not the opposite, needs to be proposed in the conclusions. We thank the reviewer for this pivotal comment.
We thus added to the Discussion the following: However, it is crucial to keep in mind that the relationship between the vagal nerve and DM may by bi-directional. Indeed, some of the experimental and prospec-tive studies reviewed above propose that the vagal nerve's functioning has a causal role in T2DM, but some of the evidence may also suggest that T2DM itself may impair vagal nerve activity, indexed by low HRV in patients with poor glycemic control. It is especially possible that neuropathy can impair vagal activity, however, since many studies conceptualize neuropathy as low HRV, other ways to measure neuropathy need to be used to examine this issue.
Reviewer 3 Report
Reviewer comments for cells-2407436
In this manuscript, the authors highlight the anti-inflammatory roles of the vagal nerve in managing type 2 diabetes mellitus (T2DM) and suggest potential directions for future research on using the vagus nerve as a treatment target for diabetes. While this commentary provides an interesting mini-review on the relationship between the vagal nerve, inflammation, and diabetes, there are several crucial and minor issues that need to be addressed, as detailed below:
1. Some literature related to the vagus nerve and diabetes appears to be absent from this manuscript. It is recommended to conduct a thorough search of relevant databases and add the details of the search process to the manuscript.
e.g.: Payne SC et al (2022) Blood glucose modulation and safety of efferent vagus nerve stimulation in a type 2 diabetic rat model. Physiological Reports 10: e15257. doi: 10.14814/phy2.15257
e.g: Yu Y, Hu L, Xu Y, Wu S, Chen Y, Zou W, Zhang M, Wang Y, Gu Y. Impact of blood glucose control on sympathetic and vagus nerve functional status in patients with type 2 diabetes mellitus. Acta Diabetol. 2020 Feb;57(2):141-150. doi: 10.1007/s00592-019-01393-8. Epub 2019 Jul 31. PMID: 31367992; PMCID: PMC6997255.
2. While the stimulation of the vagus nerve may have beneficial effects on diabetes by reducing systemic inflammation, it is important to note that low-grade inflammation is also a common cause of other chronic metabolic diseases, such as obesity, hyperlipidemia, and fatty liver. In order to gain a better understanding, it would be helpful to further elaborate on potential mechanisms beyond the regulation of the vagus nerve that may contribute to the improvement of these metabolic disorders. Otherwise, the authors should consider discussing whether the vagus nerve may play a more dominant role in regulating diabetes compared to other contributing factors.
3. When using abbreviations in a manuscript, it is important to spell out the full name of the term the first time it appears in the text to avoid confusion for readers. Such as: CDC, PKC, AGE etc.
4. Please delete the unnecessary information on figure, like created in Biorender.com
Author Response
Reviewer 3.
Thank you for your important notes and comments.
Q1. We thank this reviewer for suggesting to us to cite a few more important articles. Though we did cite several animal studies which experimentally tested effects of vagal nerve stimulation, we added the proposed articles indeed as following:
One study induced an animal model of T2DM and tested the effects of an implanted efferent vagal nerve stimulation (eVNS) on glycemia. eVNS reduced glycemia consistently, and a certain dose was even found in these rats to be most effective (Payne et al., 2022).
In the context of reviewing the evidence linking T2DM with vagal nerve activity, we also cited the study of Yu et al. (2020) proposed by reviewer 3 as following:
Finally, though not an experimental study, one study compared diabetic patients with controlled glucose with diabetic patients without controlled glucose and healthy controls, on HRV. That study found that HRV was lower in diabetic patients with poor glycemic control than in diabetic patients with good glycemic control, suggesting that a more severe illness may also result or even cause poorer vagal nerve activity (Yu et al., 2020; 46).
Q2. This is a crucial comment, and we thank the reviewer for pointing it out. We added this entire paragraph to demonstrate the comprehensive effects of the vagal nerve, beyond on reducing inflammation, which is highly relevant to diabetes:
Beyond these important anti-inflammatory effects of the vagus, this nerve has many more effects, which place it as a factor of potentially greater importance than other risk factors of diabetes. Activating the vagus helps to reduce eating in people with high craving for food (Meule et al., 2012; 49) and HRV increases with performing more physical activity such as resistance exercise, particularly in middle-aged people (Kingsley & Figuerosa, 2016; 50). This is crucial for preventing T2DM. The robust and comprehensive bi-directional effects of the vagal nerve on behavioral life-style factors (diet, exercise and smoking) and on biological mediators (oxidative stress, inflammation and sympathetic hyperactivity) have been proposed in a neuroimmunological framework of fatal diseases (Gidron et al., 2018; 50), which indeed proposes the cen-trality of this nerve in disease prediction and prevention.
Q3. Abbreviations were explained.
Q4. Redundant information was removed from the figure.
Round 2
Reviewer 1 Report
The authors made significant changes. Manuscript may be accepted in the present form
Reviewer 3 Report
The authors have appropriately revised the manuscript in accordance with the reviewer's comments.